# Simulation and Controller Design for a Fish Robot with Control Fins

**DOI:** 10.3390/biomimetics9060317

**Published:** 2024-05-25

**Authors:** Sandhyarani Gumpina, Seungyeon Lee, Jeong-Hwan Kim, Hoon Cheol Park, Taesam Kang

**Affiliations:** 1Department of Aerospace Information Engineering, Konkuk University, Seoul 05029, Republic of Korea; sandhyagumpina11@gmail.com; 2Department of Smart Vehicle Engineering, Konkuk University, Seoul 05029, Republic of Korea; sehrgut010801@gmail.com (S.L.); kjh23135@gmail.com (J.-H.K.); hcpark@konkuk.ac.kr (H.C.P.)

**Keywords:** fish robot, system identification, PID controller, six-degree-of-freedom equation

## Abstract

In this paper, a nonlinear simulation block for a fish robot was designed using MATLAB Simulink. The simulation block incorporated added masses, hydrodynamic damping forces, restoring forces, and forces and moments due to dorsal fins, pectoral fins, and caudal fins into six-degree-of-freedom equations of motion. To obtain a linearized model, we used three different nominal surge velocities (i.e., 0.2 m/s, 0.4 m/s, and 0.6 m/s). After obtaining output responses by applying pseudo-random binary signal inputs to a nonlinear model, an identification tool was used to obtain approximated linear models between inputs and outputs. Utilizing the obtained linearized models, two-degree-of-freedom proportional, integral, and derivative controllers were designed, and their characteristics were analyzed. For the 0.4 m/s nominal surge velocity models, the gain margins and phase margins of the surge, pitch, and yaw controllers were infinity and 69 degrees, 26.3 dB and 85 degrees, and infinity and 69 degrees, respectively. The bandwidths of surge, pitch, and yaw control loops were determined to be 2.3 rad/s, 0.17 rad/s, and 2.0 rad/s, respectively. Similar characteristics were observed when controllers designed for linear models were applied to the nonlinear model. When step inputs were applied to the nonlinear model, the maximum overshoot and steady-state errors were very small. It was also found that the nonlinear plant with three different nominal surge velocities could be controlled by a single controller designed for a linear model with a nominal surge velocity of 0.4 m/s. Therefore, controllers designed using linear approximation models are expected to work well with an actual nonlinear model.

## 1. Introduction

Autonomous underwater vehicles (AUVs) are designed to mimic the efficient and agile movements of fish. They are attracting attention because they can be effectively used for exploration, monitoring, and surveillance of underwater environments [1]. These AUVs integrate advanced propulsion mechanisms such as Body Caudal Fin (BCF) and Median and Pectoral Fin (MPF) systems. The BCF system mimics the natural movements of fish, which propel themselves by flapping their bodies and tails. It is used by more than 85% of fish species [2].

Several studies have been conducted on the dynamics of fish robots. Humphreys [3] derived a detailed six-degree-of-freedom (DOF) equation of motion for surge, sway, heave, roll, pitch, and yaw motions, including moments of inertia and hydrodynamic forces. Nahon [4] proposed body lift and drag forces to model the hydrodynamic characteristics of vehicles. The dynamics of an underwater vehicle, Autolycus, have been well described by Tang [5]. Vorticity control for propulsion and maneuvering was introduced by the Draper Laboratory, which allows fish to maintain stable swimming [6]. Open Fish utilizes a simple and effective wire-driven active and passive compliant body to mimic highly efficient thunniform swimming [7,8]. Tuna Bot aims to achieve high-performance fish and overcome the gap between robotic systems and fish swimming capabilities [9].

In UC-Ika 1 [10], inspired by the swimming motion of tuna fish, undulatory motions by the tail peduncle and caudal fin were used to generate propulsion force. This benefits from a tail mechanism that plays a crucial role in the dynamic behavior of the robot. A framework was developed to compute the steady-turning motion of a robotic fish undergoing periodic body or tail deformation [11]. Flying fish performing gliding flights with tail-beating motion are being developed for future applications [12,13]. Full-body-length swimming motion coordinates for anterior, midbody, and posterior displacements were proposed in iSplash-I to reduce large kinematic errors in existing free-swimming robotic fish [14]. In iSplash-II, a new fabric technique was introduced to achieve high-speed propulsion at high frequencies [15].

In previous studies [16,17], pectoral fins were used for roll and pitch controls. Fish pectoral fins can produce high propulsive performance by driving active and passive fin deformation. A systematic approach was suggested to control pectoral fins by naturally accommodating fin constraints and automatically generating “intelligent” behavior (such as fin backing-up when required) for quick maneuvering [18].

Euler’s angle speed control and 3D performance of a fish robot, including closed-loop control system responses, can be found in [19]. A two-DOF barycenter mechanism was proposed to provide body stabilization and serve as an actuating device for active control design [20]. Zeng et al. [21] proposed an underwater robot based on the hybrid propulsion of a quadrotor and undulating fin, which combines high-efficiency propulsion using a bionic undulating fin and stable control via the propeller of an underwater robot. The hybrid system enables independent heave motion, surge motion, in situ steering, and stable hovering. Kim et al. [22] proposed an integral sliding mode controller (ISMC) to stabilize an autonomous underwater vehicle (AUV) with modeling errors.

Aruna et al. [23] analyzed trajectory tracking control methods using a conventional proportional–integral–derivative (PID) control, H∞ control, and a feedforward (along with feedback) control. Xiang et al. [24] proposed a simple PID controller to drive individual AUVs following a 3D path by exploiting the 3D guidance law in the underactuated mode. Yang et al. [25] suggested the nonlinear formation keeping and mooring control of multiple autonomous underwater vehicles using the backstepping method. Qiao et al. [26] developed an effective control method to improve the trajectory tracking control in an underwater vehicle using an adaptive fast nonsingular integral terminal sliding mode control. Oktafianto et al. [27] presented a nonlinear model of an autonomous underwater vehicle (AUV) with six degrees of freedom, which was linearized using the Jacobian matrix.

In this paper, feedback control loops for a fish robot were developed using MATLAB Simulink [28]. A nonlinear model was obtained by incorporating hydrodynamic effects (damping coefficients and added masses) into conventional six-DOF free body dynamic equations.

To design a linear feedback controller, a linear model was derived from the nonlinear model using system identification tools [29]. Using the identified linear model and PID tuning, we obtained a robust PID controller with large gain and phase margins to overcome nonlinear uncertainties. The frequency responses of the linear model with a feedback controller were analyzed to check the gain margin and phase margin. After obtaining the linear model controller, we applied it to both a linearized model and a nonlinear model to compare its performances in both ideal and realistic situations. In the following sections, we will present the complete dynamic equation, the MATLAB Simulink model of the nonlinear model, linear models for surge, pitch, and yaw motions obtained from nonlinear model responses, closed-loop PID controllers for surge, pitch, and yaw motions using linear models, and the final simulation results of the controllers using nonlinear models.

## 2. Equation of Motion and Dynamics of Fins

In this section, a complete six-DOF equation of motion was derived to describe the motion of an underwater vehicle with pectoral, dorsal, and caudal fins.

### 2.1. Equation of Motion

The vehicle can move freely with six-DOF motions: surge, sway, heave, roll, pitch, and yaw motions. Figure 1 illustrates the body coordinate system of a fish robot. The origin of the body frame is at the geometrical center of the fish robot’s body. The prototype structure is a modified version of Open Fish [7,8]. The center of gravity (CG) is located below the origin. The center of buoyancy (CB) coincides with the origin of the body coordinate. In our fish robot, the buoyancy force (B) is equal to the weight (W) to maintain a neutral buoyant force condition.

Assuming the xz plane as a plane of symmetry, six-DOF equations of motion are given by Equations (1)–(6). The right-hand side of each equation represents the sum of external forces and moments arising from added masses, hydrodynamic damping, buoyancy, weight, and control. The left-hand side represents rigid body dynamics. The control forces and moments generated by the control fins are described in Section 2.2. For a complete derivation, refer to [5].
(1)$$m\left\{\dot{u}-vr+wq-x_{G}\left(q^{2}+r^{2}\right)+z_{G}\left(pr+\dot{q}\right)\right\}\;\;\;\;\;=X_{\dot{u}}\dot{u}+Z_{\dot{w}}wq+Z_{\dot{q}}q^{2}-Y_{\dot{v}}vr-Y_{\dot{r}}r^{2}+X_{u\left|u\right|}u\left|u\right|\;\;\;\;-\left(W-B\right)\sin\theta +X_{c},$$
(2)$$m\left\{\dot{v}-wp+ur+z_{G}\left(qr-\dot{p}\right)+x_{G}\left(qp+\dot{r}\right)\right\}\;\;\;\;=Y_{\dot{v}}\dot{v}+Y_{\dot{r}}\dot{r}+X_{\dot{u}}ur-Z_{\dot{w}}wp-Z_{\dot{q}}pq+Y_{v\left|v\right|}v\left|v\right|\;\;\;\;\;\;+\left(W-B\right)\cos\theta \sin\phi +Y_{c},$$
(3)$$m\left\{\dot{w}-uq+vp-z_{G}\left(p^{2}+q^{2}\right)+x_{G}\left(rp-\dot{q}\right)\right\}\;\;\;\;\;\;=Z_{\dot{w}}\dot{w}+Z_{\dot{q}}\dot{q}-X_{\dot{u}}uq+Y_{\dot{v}}vp+Y_{\dot{r}}rp+Z_{w\left|w\right|}w\left|w\right|\;\;\;\;\;\;+\left(W-B\right)\cos\theta \cos\phi +Z_{c},$$
(4)$$I_{xx}\dot{p}+\left(I_{zz}-I_{yy}\right)qr-m\left\{z_{G}\left(\dot{v}-wp+ur\right)\right\}\;\;\;\;\;\;=K_{\dot{p}}\dot{p}-\left(Y_{\dot{v}}-Z_{\dot{w}}\right)vw-\left(Y_{\dot{r}}+Z_{\dot{q}}\right)wr+\left(Y_{\dot{r}}+Z_{\dot{q}}\right)vq\;\;\;\;\;\;\;-\left(M_{\dot{q}}-N_{\dot{r}}\right)qr+K_{p\left|p\right|}p\left|p\right|-\left(z_{G}W-z_{B}B\right)\cos\theta \sin\phi +K_{c},$$
(5)$$I_{yy}\dot{q}+\left(I_{xx}-I_{zz}\right)rp+m\left\{z_{G}\left(\dot{u}-vr+wq\right)-x_{G}\left(\dot{w}-uq+vp\right)\right\}\;\;\;\;=Z_{\dot{q}}\left(\dot{w}-uq\right)+M_{\dot{q}}\dot{q}-\left(Z_{\dot{w}}-X_{\dot{u}}\right)wu-Y_{\dot{r}}vp+\left(K_{\dot{p}}-N_{\dot{r}}\right)rp\;\;\;\;\;+M_{q\left|q\right|}q\left|q\right|+M_{w\left|w\right|}w\left|w\right|-\left(z_{G}W-z_{B}B\right)\sin\theta \;\;\;-x_{G}\left(W-B\right)\cos\theta \cos\phi +M_{c},$$
(6)$$I_{zz}\dot{r}+\left(I_{yy}-I_{xx}\right)pq+m\left\{x_{G}\left(\dot{v}-wp+ur\right)\right\}\;\;\;\;\;\;\;=Y_{\dot{r}}\dot{v}+N_{\dot{r}}\dot{r}-\left(X_{\dot{u}}-Y_{\dot{v}}\right)uv+Y_{\dot{r}}ur+Z_{\dot{q}}wp-\left(K_{\dot{p}}-M_{\dot{q}}\right)pq\;\;\;\;\;+N_{r\left|r\right|}r\left|r\right|+N_{v\left|v\right|}v\left|v\right|+x_{G}\left(W-B\right)\cos\theta \sin\phi +N_{c}.$$

Variables u, v, w, p, q, r, ϕ, and θ indicate surge velocity, sway velocity, heave velocity, roll rate, pitch rate, yaw rate, roll angle, and pitch angle, respectively. Body mass and the moment of inertia along the x, y, and z axes are *m*, $I_{xx},\;I_{yy},$ and ${\;I}_{zz}$, respectively. The positions of the center of buoyancy and center of gravity are $\left(x_{B},y_{B},z_{B}\right)$ and $\left(x_{G},y_{G},z_{G}\right)$, respectively. 

The added masses corresponding to surge, sway, heave, roll, pitch, and yaw motions are denoted as $X_{\dot{u}},{\;Y}_{\dot{v}},{\;Z}_{\dot{w}},{\;K\;}_{\dot{p}},{\;M}_{\dot{q}},{\;N}_{\dot{r}},\;Y_{\dot{r}},$ and $Z_{\dot{q}}$, respectively. The ellipsoidal body is assumed to estimate the added mass and moments of inertia of the vehicle [30,31]. The effects of the control fins were neglected. Assuming a symmetric body, we have $N_{\dot{v}}=Y_{\dot{r}}$, $M_{\dot{w}}=Z_{\dot{q}}$. 

The damping forces are represented by $X_{u\left|u\right|}u\left|u\right|$, $Y_{v\left|v\right|}v\left|v\right|$, $Z_{w\left|w\right|}w\left|w\right|$, $K_{p\left|p\right|}p\left|p\right|$, $M_{w\left|w\right|}w\left|w\right|$, $M_{q\left|q\right|}q\left|q\right|$, $N_{r\left|r\right|}r\left|r\right|$, and $N_{v\left|v\right|}v\left|v\right|$, illustrating the nonlinear and coupled damping characteristics of an underwater vehicle in six-DOF motions at a high speed [5,6,7,8,9,10,11,12,13,14,15,16,17,18,19,20,21,22,23,24,25,26,27,28,29,30,31,32,33]. The forces and moments due to the control fins (i.e., $X_{c},Y_{c},Z_{c},K_{c},M_{c},$ and $N_{c}$) will be detailed in Section 2.2.

The relationship between the Euler angle and the body axis rate is given as Equations (7)–(9) [5]. The yaw angle is denoted as ψ.
(7)$$\dot{\phi }=p+q\sin\phi \tan\theta +r\cos\phi \tan\theta ,$$
(8)$$\dot{\theta }=q\cos\phi -r\sin\phi ,$$
(9)$$\dot{\psi }=q\sin\phi \sec\theta +r\cos\phi \sec\theta .$$

Table 1 and Table 2 show parameters used in the control simulations. They were estimated based on the size and shape of the body. The damping coefficients were approximated assuming a flat ellipsoidal body. To be more realistic, a more accurate calculation of these parameters and verification via real experiments are necessary. 

### 2.2. Forces and Moments Due to Control Fins

Caudal fins were utilized to generate thrust, whereas pectoral and dorsal fins were used to control the pitch and yaw motions of the fish robot. The caudal fin produced the propulsion force necessary for propelling the fish forward. We set the tail beat angle to 50 degrees and estimated the average thrust produced by the tail when the flapping frequency varied from 1 Hz to 7 Hz. The surge force ($X_{c}$) can be expressed using Equation (10). The terms $T_{cd}$, $D_{b}$, $D_{pf}$, and $D_{df}$ represent thrust by the caudal fin, body drag, pectoral fin drag, and dorsal fin drag, respectively.
(10)$$X_{c}=T_{cd}-D_{b}-D_{pf}-D_{df}.$$

Figure 2 shows thrusts generated by the caudal fin. These thrusts were calculated using the added mass method [12,34,35], first proposed by Lighthill [35] and then modified for calculating the thrust generated by a forked-shaped caudal fin [12]. The fin used in [12] was replaced by the current fin geometry. The driving frequency varied from 1 to 7 Hz for the thrust calculation, as shown in Figure 2.

To calculate the drag and lift forces exerted by the fins, we used the general drag force ($F_{d}$) equation in [4], given by Equations (11)–(13):(11)$$F_{d}=\frac{1}{2}\rho U^{2}AC_{d},$$
where U is the forward velocity, A is the relevant reference area of the fin, and $C_{d}$ is the drag coefficient given in Equation (12). In Equation (12), $C_{D_{o}}$ is the parasite drag, $A_{r}$ is the aspect ratio of a fin, e is the Oswald efficiency factor, which typically has a value of 0.85 to 0.9 for any airfoil section, and $C_{l}$ is the coefficient of the lift.
(12)$$C_{d}=C_{D_{o}}+\frac{C_{l}^{2}}{\pi A_{r}e},$$

To estimate the coefficient of the lift, we referred to Equation (13), where $C_{l\alpha }$ is the slope of the curve with the angle of attack α. We assumed that fin shapes were roughly similar to the shape of the NACA0012 airfoil [36]. $C_{l\alpha }$ is obtained by assuming that the maximum α is 15 degrees.
(13)$$C_{l}=C_{l\alpha }\frac{A_{r}}{A_{r}+\left[\frac{2(A_{r}+4)}{\left(A_{r}+2\right)}\right]}\alpha ,$$

Based on Equations (11)–(13), we calculated the drag forces of the pectoral and dorsal fins. For the body drag, we used Equation (11), assuming a drag coefficient $C_{d}\;$of 0.032, which corresponded to the wetted surface area of a fish robot [37].

To control the pitch moment in fish robots, we drove pectoral fins so that the angle of attack was between −15~+15 degrees. The heave force $Z_{cpf}$ and pitch moment $M_{cpf}$ of the pectoral fins are given by Equations (14) and (15):(14)$$Z_{c}=Z_{cpf}=-L_{pf},$$
(15)$$M_{cpf}=L_{pf}\left(x_{p}\right)-D_{pf}\left(z_{p}\right).$$

In Equations (14) and (15), $L_{pf}$ represents combined lift forces from both left and right pectoral fins, as shown in Figure 3a. The position of the pectoral fin was denoted as ($x_{p}{,y}_{p},z_{p}$). $D_{pf}$ denotes the drag force exerted by the left and right pectoral fins. During the pitching motion, heave velocity is induced by the lift force, as shown in Equation (14). The lift force ($F_{l}$) of a fin is given by Equation (16):(16)$$F_{l}=\frac{1}{2}\rho U^{2}AC_{l}.$$

The lift and drag forces of the dorsal fin are shown in Figure 3b. Force and moments by the dorsal fin are given by Equations (19) and (20):(17)$${Y_{c}=Y}_{cdf}=L_{df},$$
(18)$$K_{c}=K_{cdf}=-L_{df}\left(z_{d}\right),$$
(19)$$M_{cdf}=-D_{df}(z_{d}),$$
(20)$${N_{c}=N}_{cdf}=L_{df}\left(x_{d}\right).$$

In Equation (17), the lift force exerted by the dorsal fin, $L_{df},$ also works for the sway directional acceleration force $Y_{cdf}$. $K_{cdf}$ in Equation (18) represents the roll moment by the dorsal fin. The position of the dorsal fin is represented by ($x_{d},y_{d},\;z_{d}$). $M_{cdf}$ in Equation (19) denotes the pitch moment resulting from the drag force of the dorsal fin. $N_{cdf}$ in Equation (20) represents the yaw moment by the dorsal fin. Combining Equations (15) and (19), we obtained the pitch control moment for the pectoral and dorsal fins, as shown in Equation (21).
(21)$${M_{c}=M_{cpf}+M}_{cdf}=L_{pf}\left(x_{p}\right)-D_{pf}\left(z_{p}\right)+D_{df}\left(z_{d}\right)$$

## 3. Simulation and Controller Design Results

We initially conducted an open-loop simulation for our robotic fish, where control inputs were applied by caudal, pectoral, and dorsal fins using Equations (10)–(21). Figure 4 shows a simplified simulation block for a fish robot. The outputs measured were surge, sway, and heave velocities, as well as roll, pitch, and yaw rates, in addition to roll, pitch, and yaw angles.

As shown in Equation (22), two-DOF PID controllers were used for surge, pitch, and yaw controllers.
(22)$$G(s)=P(br-y)+I\frac{1}{s}(r-y)+D\frac{N}{1+N\frac{1}{s}}(cr-y),$$
where P, I, D, and N are the proportional gain, integral gain, derivative gain, and derivative filter time, respectively. b and c are the set point weights on the proportional term and derivative term. r is the reference input, and y is the output. G(s) is the controller transfer function, and s is the complex frequency in the Laplace transform [38]. For practical use, the control loop [39,40,41,42] was designed to satisfy the following preferred conditions: (1) the gain margin should be higher than 10 dB to cover the model uncertainties [43]; and (2) the phase margin should be larger than 45° to cover time delays and model uncertainties [44,45]. 

### 3.1. Surge Response and Velocity Controller Design

To obtain surge responses, the nominal velocity commands were given as 0.2 m/s (Condition 1), 0.4 m/s (Condition 2), and 0.6 m/s (Condition 3). At 10 s, after a steady-state surge velocity was reached, a pseudo-random binary signal (PRBS) control input of ±0.05 m/s was added to the nominal input to check the plant response variations, as shown in Figure 5a. In Figure 5a, the nominal velocities were subtracted to show the velocity variations only. Input and output data were used to obtain transfer functions between velocity command variations and output velocity variations. After obtaining surge open-loop responses from the nonlinear model, we applied the system identification tool in MATLAB to obtain a linearized transfer function. Figure 5b–d show verifications of the identified linear model using different PRBS input signals. The output response of the identified linear model and that of the original nonlinear model were compared. The fits to estimations with nominal velocities of 0.2 m/s, 0.4 m/s, and 0.6 m/s were 61.9%, 91.3%, and 92.8%, respectively. Table 3 shows the linearly approximated transfer functions obtained when the nominal surge velocities were 0.2 m/s, 0.4 m/s, and 0.6 m/s. Figure 5e shows the frequency responses of linearly identified plants. The low-frequency gains decreased as the nominal surge velocity decreased. As the nominal surge velocity increased, the loop gains and bandwidths also increased. Thus, the responses to the control inputs were faster.

Two-DOF PID controllers, as shown in Equation (22), were designed for the model transfer functions in Table 3. Table 4 shows the surge controller parameters obtained using two-DOF PID tuning.

Figure 5f shows the frequency responses of plants and PID controllers to check system stability margins. As shown in Table 5, the bandwidths of the three models were 0.90 rad/s, 2.32 rad/s, and 2.19 rad/s, and the phase margins were 69 degrees, 69 degrees, and 70 degrees, respectively. The gain margin was infinite. Figure 5g shows the step responses of a closed-loop linearized plant with the obtained PID controllers. The rise times were about 4, 2, and 2 s. Both the maximum overshoot and steady-state error were small enough to be ignored. Figure 5h shows the step responses of closed-loop nonlinear plant responses when linearly obtained PID controllers are applied. The results showed that PID controllers work well even when they are applied to nonlinear models. 

Figure 6 shows the step responses of the closed-loop nonlinear plant when the linear controller designed using a nominal velocity of 0.4 m/s is applied to nonlinear plants with different nominal velocities. The results showed that the PID controller, originally optimized for 0.4 m/s, performs effectively with different nominal velocities. This robust performance is due to the fact that the controller has large gain and phase margins, which ensure stability and responsiveness under varying conditions.

### 3.2. Pitch Response and Pitch Controller Design

To obtain the pitch response, a PRBS signal of ±0.5 radians was used as the pitch input. Surge velocities were set to 0.2 m/s, 0.4 m/s, and 0.6 m/s. Initially, an open-loop simulation for the pitch was conducted, and input–output data were collected to identify the plant model, as shown in Figure 7a. From the resulting input–output responses, we derived the linearized transfer function using the MATLAB Simulink System Identification Tool. As shown in Figure 7b–d, the nonlinear and linear model responses match well with another PRBS input. The fits to estimations for 0.2 m/s, 0.4 m/s, and 0.6 m/s were 86.14%, 84.3%, and 86.14%, respectively. 

The linear transfer functions obtained are shown in Table 6. Figure 7e shows the frequency responses of the obtained linear models. 

A two-DOF PID controller was designed using PID tuning for the linear transfer function, as shown in Table 6. The parameters of the two-DOF PID controller obtained are shown in Table 7.

Figure 7f shows the frequency responses of the linear pitch plant models with feedback controllers to check the system’s performances. Table 8 shows the gain and phase margins and bandwidths of the closed-loop control systems. The gain margins are large enough, even though they are not infinite, as in the surge control. The phase margins are larger than 76 degrees, which is more than enough. Figure 7g shows the step responses of closed-loop pitch controllers for linear plant models with different nominal surge velocity conditions. The maximum rise time was about 20 s. Both maximum overshoots and steady-state errors were negligibly small. Figure 7h shows step responses when the PID controllers derived from linear models are applied to the nonlinear model. From the start to 10 s, pitch reference inputs remained at zero. Due to the surge velocities being changed from zero to nominal velocities, there were some oscillations in the pitch angle. At 10 s, 0.01-rad pitch commands were applied to the pitch controllers. It could be seen that all pitch-angle outputs were following the commands. The rise time was about 5 s, demonstrating that the obtained PID controller also performs well with the nonlinear plant. 

Figure 8 shows the step response of a nonlinear plant when the linearly derived pitch controller for the 0.4 m/s model is applied with different nominal surge velocities. When the nominal surge velocities are 0.2 m/s, 0.4 m/s, and 0.6 m/s, the rise time is about 15 s, 2 s, and 1 s, respectively. When the surge velocity was low, it took more time to maneuver the pitch because the pitching force is proportional to the square of the surge velocity.

### 3.3. Yaw Response and Yaw Controller Design

In the yaw control loop simulation, the surge nominal velocities were given as 0.2 m/s, 0.4 m/s, and 0.6 m/s. The PRBS input with an amplitude of 0.5 rad was given as a yaw control input after 10 s, as shown in Figure 9a. After applying the system identification tool in MATLAB Simulink to the yaw command input and output data, we obtained a linear transfer function describing characteristics between the yaw command input and output. The obtained linear transfer functions are shown in Table 9. Figure 9b, Figure 9c, and Figure 9d show the responses of the identified linear and nonlinear models. For the 0.2 m/s, 0.4 m/s, and 0.6 m/s models, the fits to estimations were 72.59%, 64.09%, and 64.37%, respectively. Figure 9e shows the frequency responses of the obtained linear models.

Two-DOF PID controllers were designed and tuned in Simulink using transfer functions obtained from the system identification, as shown in Table 9. The parameters of these two-DOF PID controllers are shown in Table 10.

Figure 9f shows the frequency responses of the linearized yaw plant with feedback controllers. Table 11 shows the gain and phase margins for the obtained controller. Figure 9g shows the closed-loop responses of the linearized models with the obtained controller. The results showed that for yaw control systems with surge velocities of 0.2 m/s, 0.4 m/s, and 0.6 m/s, the rise times were 2 s, 2.5 s, and 3 s, respectively. The maximum overshot and steady-state error were small when the nominal surge velocities were 0.4 m/s and 0.6 m/s. When the nominal surge velocity was 0.2 m/s, the output had some overshoot and slowly reached a steady state. Figure 9h shows the step responses of a nonlinear plant with linearly obtained PID controllers. At surge velocities of 0.2 m/s, 0.4 m/s, and 0.6 m/s, the rise times were 2 s, 2.7 s, and 5 s, respectively. 

Figure 10 shows the step responses of the linearly derived yaw controller with 0.4 m/s nominal surge velocity applied to the nonlinear model with different nominal surge velocities. When the nominal surge velocities were 0.2 m/s, 0.4 m/s, and 0.6 m/s, the rise time was about 4.5 s, 2.5 s, and 2.5 s, respectively. It shows that the applied controller worked well even with different conditions.

## 4. Conclusions

In this paper, a simulation block for a fish robot was constructed using MATLAB Simulink. Linearly approximated transfer function models were obtained from surge, pitch, and yaw open-loop responses. Using these linear models, we designed a simple two-DOF PID controller for surge, pitch, and yaw control. A frequency analysis using Bode diagrams for the surge, pitch, and yaw controllers showed that the gain margin and phase margin were sufficiently large for the surge, yaw, and pitch controllers. 

When surge, pitch, and yaw controllers were applied to the linear model, the maximum overshoot and steady-state error were small. The bandwidths of the surge pitch and yaw controllers were 2.3 rad/s, 0.17 rad/s, and 2 rad/s, respectively. When these controllers were applied directly to a nonlinear plant, the response characteristics were similar to those when the controllers were applied to a linear model. Therefore, it is expected that the proposed controller can be applied to control real fish robots with nonlinear coupling terms and uncertainties.

In simulations, we assumed a symmetric ellipsoidal body and obtained parameters simplifying the structure. Thus, these parameters need to be updated using more accurate methods and actual experimental data. In this research, we focused on single-input and single-output responses. However, in future work, we will consider multi-input and multi-output responses to see more complex system behaviors. Additionally, the current control law does not fully address interactions between surges and other forces, which could lead to instability. 

In the future, we plan to fabricate an actual fish robot. With a real fish robot, we will perform experiments to identify dynamic parameters and transfer characteristics and implement controllers to verify their performances.

## Figures and Tables

**Figure 1 biomimetics-09-00317-f001:**
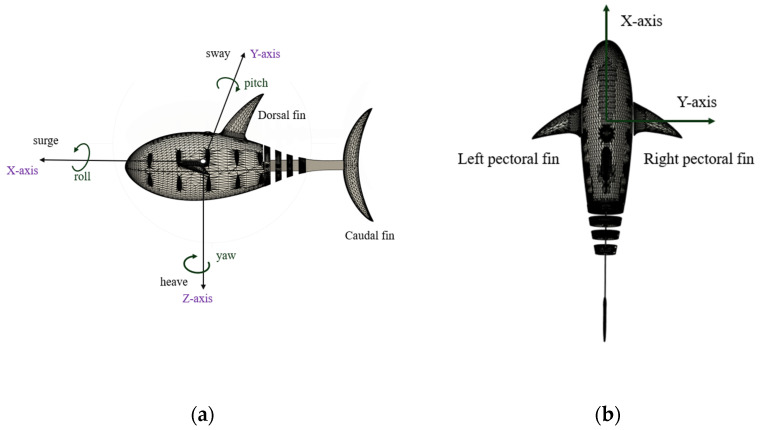
Fish robot coordinate system: (**a**) side view of the fish robot; (**b**) top view of the fish robot.

**Figure 2 biomimetics-09-00317-f002:**
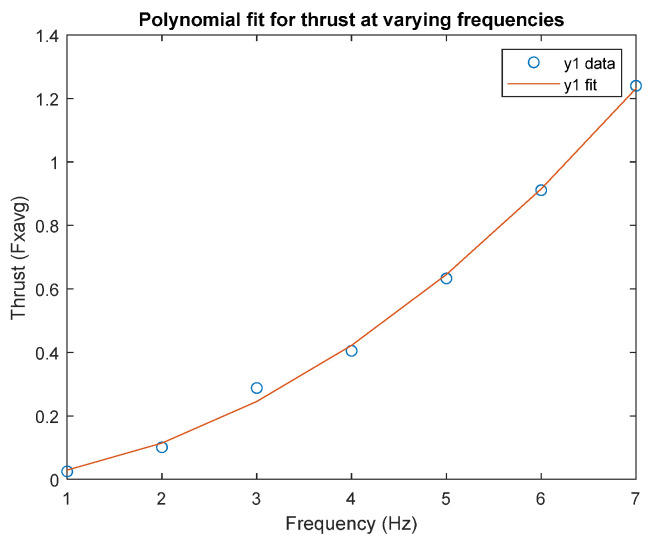
Thrust versus frequency plot.

**Figure 3 biomimetics-09-00317-f003:**
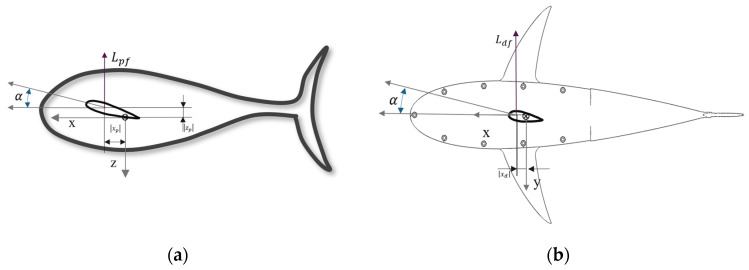
Pectoral and dorsal moments and forces: (**a**) side view of the fish robot; (**b**) top view of the fish robot.

**Figure 4 biomimetics-09-00317-f004:**
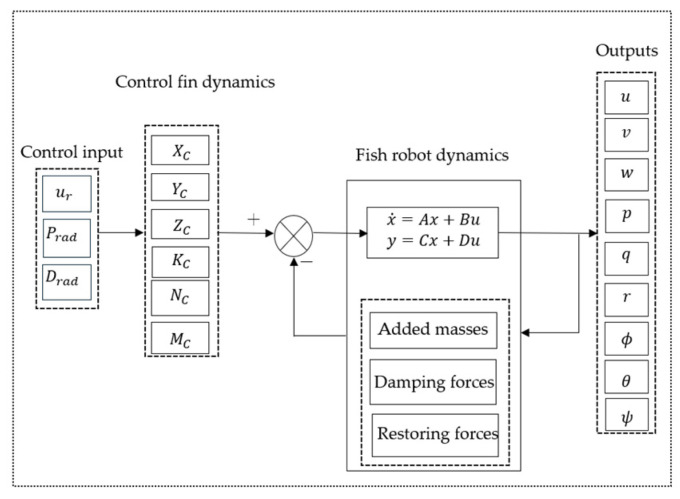
Simplified open-loop simulation block.

**Figure 5 biomimetics-09-00317-f005:**
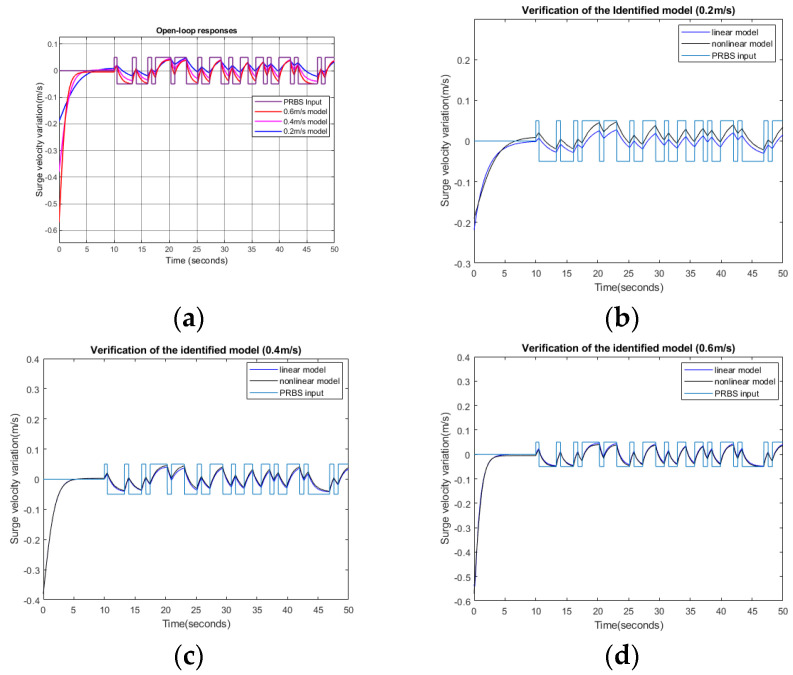
Surge velocity responses: (**a**) open-loop responses to speed commands with different nominal velocities; (**b**) responses of the identified linear model and nonlinear plant at a nominal speed of 0.2 m/s; (**c**) responses of the identified linear model and nonlinear plant at a nominal speed of 0.4 m/s; (**d**) responses of the identified linear model and nonlinear plant at a nominal speed of 0.6 m/s; (**e**) frequency response of identified linear models; (**f**) Bode plots of identified plants with feedback controllers; (**g**) step responses of the closed-loop linearized models; (**h**) step responses of the closed-loop nonlinear model.

**Figure 6 biomimetics-09-00317-f006:**
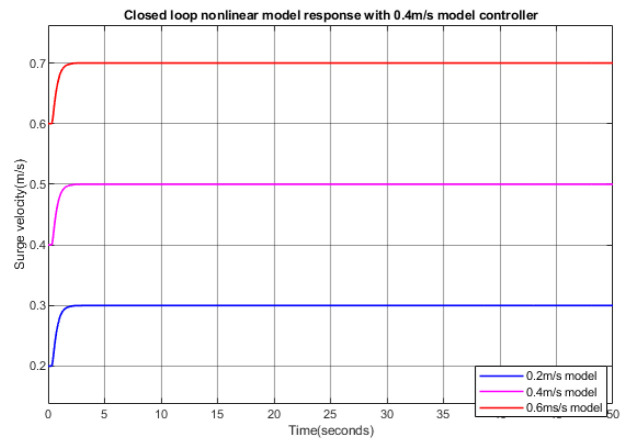
Step responses of the closed-loop nonlinear surge model with a 0.4 m/s PID controller.

**Figure 7 biomimetics-09-00317-f007:**
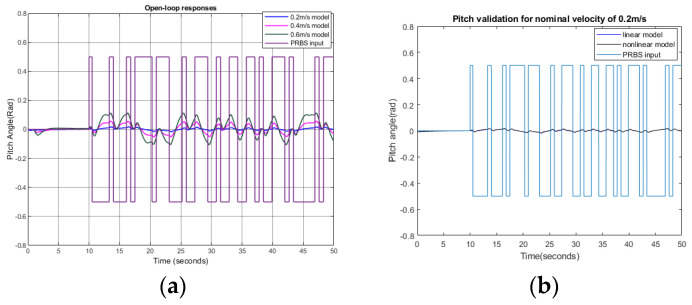
Pitch responses: (**a**) open-loop pitch responses at surge velocities of 0.2 m/s, 0.4 m/s, and 0.6 m/s; (**b**) responses of the identified linear model and nonlinear plant at a nominal speed of 0.2 m/s; (**c**) responses of the identified linear model and nonlinear plant at a nominal speed of 0.4 m/s; (**d**) responses of the identified linear model and nonlinear plant at a nominal speed of 0.6 m/s; (**e**) frequency response of the identified linear models; (**f**) Bode plots of identified plants with feedback controllers; (**g**) step responses of closed-loop controllers with linearized models; (**h**) step responses of closed-loop controllers with a nonlinear model.

**Figure 8 biomimetics-09-00317-f008:**
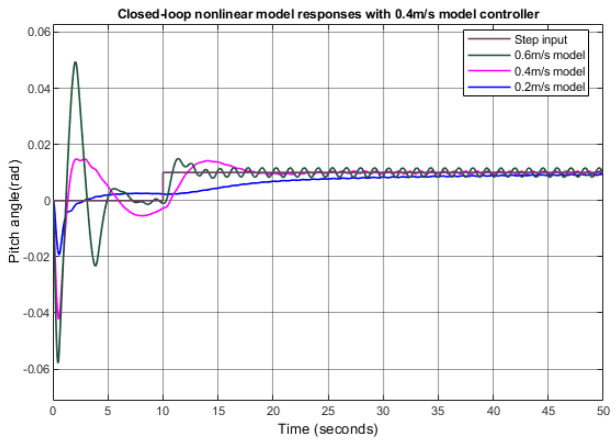
Step responses of the nonlinear plant applied with a linearly obtained pitch controller of 0.4 m/s.

**Figure 9 biomimetics-09-00317-f009:**
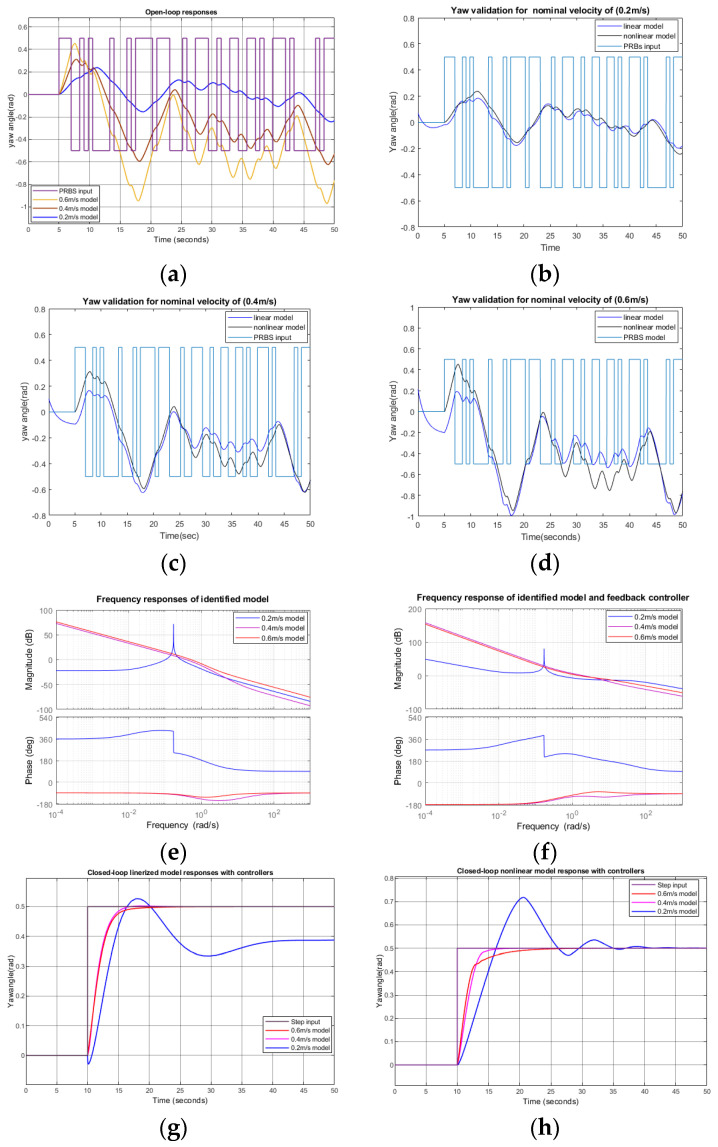
Yaw simulation responses: (**a**) open-loop yaw angle responses when PRBS yaw command input is applied; (**b**) responses of the identified linear model and nonlinear plant when the nominal velocity is 0.2 m/s; (**c**) responses of the identified linear model and nonlinear plant when the nominal velocity is 0.4 m/s; (**d**) responses of the identified linear model and nonlinear plant when the nominal velocity is 0.6 m/s; (**e**) frequency responses of the identified linear models; (**f**) Bode plots of the identified linear models with feedback controllers; (**g**) step responses of the closed-loop linearized model with PID controllers; (**h**) step responses of the closed-loop nonlinear model with linearized PID controllers.

**Figure 10 biomimetics-09-00317-f010:**
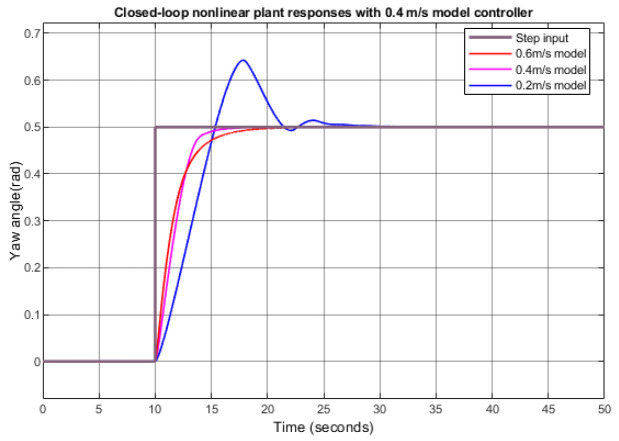
Step responses of the closed-loop nonlinear model with a 0.4 m/s yaw controller.

**Table 1 biomimetics-09-00317-t001:** Parameters of the fish robot.

Parameter	Symbol	Value	Unit
Length	l	$5.00\;\times \;10^{-1}$	m
Weight	w	$1.96\;\times \;10^{1}$	N
Buoyancy	B	$1.96\;\times \;10^{1}$	N
Moment of inertia in x-axis	$I_{xx}$	$1.30\;\times \;10^{-3}$	$kg/m^{2}$
Moment of inertia in y-axis	$I_{yy}$	$1.23\;\times \;10^{-2}$	$kg/m^{2}$
Moment of inertia in z-axis	$I_{zz}$	$1.23\;\times \;10^{-2}$	$kg/m^{2}$
Position of CB	$\left(x_{B},y_{B},z_{B}\right)$	(0,0,0)	m
Position of CG	$\left(x_{G},y_{G},z_{G}\right)$	$(0,0,\;4.00\;\times \;10^{-2}$)	m

**Table 2 biomimetics-09-00317-t002:** Hydrodynamic parameters of the fish robot.

Added Mass	Nonlinear Damping Coefficients
$X_{\dot{u}}$ $\;=-1.00\;\times \;10^{-1}\;kg$	$X_{u\left|u\right|}$ $\;=-9.30\;\times \;10^{-1}$
$Y_{\dot{v}}$ $\;=-1.40\;\times \;10^{-1}\;kg$	$Y_{v\left|v\right|}$ $\;=-3.43\;\times \;10^{1}$
$Z_{\dot{w}}$ $\;=-1.40\;\times \;10^{-1}\;kg$	$Z_{w\left|w\right|}$ $\;=-3.43\;\times \;10^{1}$
$K_{\dot{p}}$ $\;=-1.10\;\times \;10^{-2}\;kg\;m^{2}$	$K_{p\left|p\right|}$ $\;=-1.00\;\times \;10^{-1}$
$M_{\dot{q}}$ $\;=-3.00\;\times \;10^{-3}\;kg\;m^{2}$	$M_{q\left|q\right|}$ = −2.77
$N_{\dot{r}}$ $\;=-3.00\;\times \;10^{-3}\;kg\;m^{2}$	$N_{r\left|r\right|}$ = −2.77
$Y_{\dot{r}}$$\;=1.00\;\times \;10^{-3}$ kg m	$Y_{r\left|r\right|}=$ 0
$Z_{\dot{q}}$$\;=-1.00\;\times \;10^{-3}$ kg m	$M_{w\left|w\right|}=0$

**Table 3 biomimetics-09-00317-t003:** Linearly obtained surge velocity transfer functions with different nominal surge velocities.

Nominal Caudal Frequency (Hz)	Nominal Surge Velocity Condition	Linearly Obtained Transfer Function
2.00	Condition 1 (0.2 m/s)	$\frac{3.98\;\times \;10^{-1}}{s\;+\;5.08\;\times \;10^{-1}}$
4.00	Condition 2 (0.4 m/s)	$\frac{8.72\;\times \;10^{-1}\;s\;+\;8.29\;\times \;10^{-1}}{s^{2}\;+\;1.84\;s\;+\;1.09}$
6.00	Condition 3 (0.6 m/s)	$\frac{1.73\;s\;+\;1.08\;\times \;10^{1}}{s^{2}\;+\;1.02\;\times \;10^{1}\;s\;+\;1.11\;\times \;10^{1}}$

**Table 4 biomimetics-09-00317-t004:** Parameters of two-DOF PID surge controllers.

Model	P	I	D	N	b	c
Condition 1	$1.70\;\times \;10^{1}$	$1.73\;\times \;10^{1}$	−1.22	1.92	$3.02\;\times \;10^{-2}$	$2.20\;\times \;10^{-1}$
Condition 2	$2.03\;\times \;10^{1}$	$4.32\;\times \;10^{1}$	−2.80	2.75	$1.38\;\times \;10^{-2}$	$3.60\;\times \;10^{-1}$
Condition 3	$1.22\;\times \;10^{1}$	3.22	$-9.58\;\times \;10^{-1}$	5.65	$3.94\;\times \;10^{-2}$	$8.93\;\times \;10^{-2}$

**Table 5 biomimetics-09-00317-t005:** Gain and phase margins for surge controllers.

Model	Gain Margin (Decibels)	Phase Margin (Degrees)
Condition 1	Infinity	69 at 0.90 rad/s
Condition 2	Infinity	69 at 2.32 rad/s
Condition 3	Infinity	70 at 2.19 rad/s

**Table 6 biomimetics-09-00317-t006:** Linearly obtained pitch transfer functions.

Nominal Surge Condition	Linearly Obtained Transfer Function
Condition 1	$\frac{-3.00\;\times \;10^{-3}\;s^{2}\;+\;4.55\;\times \;10^{-1}\;s\;-\;9.56\;\times \;10^{-1}}{s^{3}\;+\;6.12\;s^{2}\;+\;6.24\;\times \;10^{1}\;s\;+\;2.23\;\times \;10^{1}}$
Condition 2	$\frac{3.49\;\times \;10^{-2}\;s^{2}\;+\;5.84\;\times \;10^{-1}\;s\;-\;2.8}{s^{3}\;+\;8.40\;s^{2}\;+\;3.23\;\times \;10^{1}\;s\;+\;2.65\;\times \;10^{1}}$
Condition 3	$\frac{4.27\;\times \;10^{-2}{\;s}^{2}\;+\;6.88\;\times \;10^{1}\;s\;-\;5.09}{s^{3}\;+\;8.59{\;s}^{2}\;+\;2.50\;\times \;10^{1}\;s\;+\;2.19\;\times \;10^{1}}$

**Table 7 biomimetics-09-00317-t007:** Parameters of two-DOF PID pitch controllers.

Model	P	I	D	N	b	c
Condition 1	−2.41	−1.89	0	$1.00\;\times \;10^{2}$	1	1
Condition 2	0	−0.99	0	$1.00\;\times \;10^{2}$	1	1
Condition 3	−3.07	−1.78	0	$1.00\;\times \;10^{2}$	1	1

**Table 8 biomimetics-09-00317-t008:** Gain and phase margins for pitch controllers.

Model	Gain Margin (Decibels)	Phase Margin (Degrees)
Condition 1	32.1 at 3.53 rad/s	81 at 0.07 rad/s
Condition 2	26.3 at 2.88 rad/s	85 at 0.17 rad/s
Condition 3	22.2 at 1.88 rad/s	76 at 0.23 rad/s

**Table 9 biomimetics-09-00317-t009:** Linearly obtained yaw transfer functions.

Nominal Surge Condition	Linearly Obtained Transfer Function
Condition 1	$\frac{-6.39\;\times \;10^{-2}\;s^{2}\;+\;1.17\;\times \;10^{-1}\;s\;+\;1.48\;\times \;10^{-3}}{s^{3}\;+\;6.38\;\times \;10^{-1}\;s^{2}\;+\;2.93\;\times \;10^{-2}\;s\;+\;1.87\;\times \;10^{-2}}$
Condition 2	$\frac{2.97\;\times \;10^{-2}\;+\;3.07\;\times \;10^{-1}}{s^{2}\;+\;6.92\;\times \;10^{-1}\;s\;+\;2.88\;\times \;10^{-10}}$
Condition 3	$\frac{1.66\;\times \;10^{-1}\;s\;+\;4.48\;\times \;10^{-1}}{s^{2}\;+\;6.96\;\times \;10^{-1}\;s\;+\;5.469\;\times \;10^{-11}}$

**Table 10 biomimetics-09-00317-t010:** Parameters of two-DOF PID yaw controllers.

Model	P	I	D	N	b	c
Condition 1	2.20	0.35	3.40	$4.93\times 10^{-1}$	$8.96\;\times \;10^{-1}$	$2.15\;\times \;10^{-1}$
Condition 2	6.57	1.72	5.61	5.41	$5.30\;\times \;10^{-1}$	2.00
Condition 3	3.24	0.72	2.09	5.58	$5.69\;\times \;10^{-1}$	2.00

**Table 11 biomimetics-09-00317-t011:** Gain and phase margins for yaw controllers.

Model	Gain Margin (Decibels)	Phase Margin (Degrees)
Condition 1	13.0 at 9.15 rad/s	57 at 0.43 rad/s
Condition 2	Infinity	69 at 2.00 rad/s
Condition 3	Infinity	85 at 1.52 rad/s

## Data Availability

The datasets presented in this article are not readily available because of an ongoing study.

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
