# Peer review of "Simulation and Controller Design for a Fish Robot with Control Fins"

_biomimetics, 2024, doi:10.3390/biomimetics9060317_

Round 1

Reviewer 1 Report

Comments and Suggestions for Authors

Overall I think the idea is interesting and novel. However, I think the execution left a lot to be desired. 

1. The introduction lacks sufficient references and is poorly organized. I was shocked that the first two paragraphs did not include any references. This made all the claims very weak. Citing a few review articles may be helpful here. And the sentence that "it is used by 85% of the fish species" definitely needs reference to back up.

2. The weakest part of the manuscript is its method section. Critically, many important pieces of the project were left out! Line 124 mentioned a CFD analysis, but no further details were provided. Many parameters used in this study, such as zb, zwere not included in Table 1. There also lacked discussions about the method of system identification. 

3. There were also a few key flaws in this project:

3.1 The dynamic equations (1-6) seem to ignore a few important aspects. Where is the origin of the robot body coordinate defined? Why does the center of buoyancy only have the z component zb and no xb? Simulink should have a rigid-body dynamics model, why must the authors use equations that may be error-prone? 

3.2 Why is the buoyancy equal to the weight in the surge simulations and not the others? This is certainly not the case for real robots and should not be a criterion for successfully controlling the vehicle.

3.3 How is the linear model obtained? What data did it use? If only one experiment of the surge response was used, is the linear model just a fit of the response? I think for each case (surge, pitch yaw), there is a lot of symmetry that can be exploited if the authors just canceled terms from eq (1-6), they would have also ended up with a lower-ordered model that is more realistic and useful. Regardless, assuming symmetry and reducing the dimension means that the control strategy discovered can only be used when these conditions are satisfied. That is to say, the surge control that the authors have found may only be used when the vehicle has no orientational velocities and no velocity in the heave and sway direction.

3.4 Fig 6(a) was extremely puzzling. Was the 6 kinks for the 6 control input? To do this correctly, the authors should allow the system to reach equilibrium for each control input (thrust).

3.5 Key variables Gs, s, r, y, were not defined at all and the numbers presented from eq 22-27 read unprofessional as the numbers have different significant digits.

Overall, I am not convinced that the control discovered in this project can be used in underwater vehicles.

Comments on the Quality of English Language

Line 18: The grammar is off.

Line79: "In this section" instead of "this section"

Author Response

Dear editor and reviewers

Thank you for the valuable comments from the editor and reviewers. We worked very hard to accommodate all comments in a short period of time, and as a result, the quality of the paper improved significantly, for which we are grateful.

We have thoroughly checked the English expressions, but due to lack of time, we have not yet been able to receive expert help. However, in the future, we will improve the quality of our English expressions with the help of native speaking experts.

The authors' responses to each of the reviewers' comments are attached as separate attachments. Additionally, the corrected parts of the paper are written in red to make them easier to read.

We would like to again express our sincere gratitude to the reviewers for their efforts in improving our manuscript.

Reviewer 2 Report

Comments and Suggestions for Authors

This paper designs a nonlinear simulation block for a fish robot in MATLAB Simulink, integrating various hydrodynamic forces and fins. PID controllers, designed from linearized models, exhibit strong performance with minimal overshoot and steady-state errors when applied to the nonlinear model, suggesting effective real-world application. The subject matter is intriguing, yet there are minor suggestions for improvement. The specific points to address are outlined below:

Clearly articulate the novelty of the proposed control algorithm in the abstract. It remains ambiguous which aspect of the design is innovative.

The derivation of the control design lacks sufficient explanation. Clarification is needed regarding how the control law was derived, including the steps involved.

Justification for the selection of controller gains is necessary.

Provide further elucidation on the advantages and enhancements of the proposed method and technology. Additionally, compare these findings with existing literature. Acknowledge that the control design techniques employed here are akin to those in other studies, but highlight and discuss the challenges encountered in this research to demonstrate that it is not merely an incremental extension of existing methodologies.

I understand that hardware realization may not be possible due to lack of hardware resources. However, please include a critical discussion on what could be anticipated challenges if the proposed algorithm is realized on a real quadrotor system.

Integrating additional performance indices could offer a more comprehensive verification of the controller's performance.

Enhance the discussion on existing control algorithms in the introduction with recent references, such as those available at https://doi.org/10.1016/j.ins.2023.120087, doi.org/10.1016/j.isatra.2021.02.045.

Incorporate a discussion on the limitations of the control law in the conclusions section.

Address the issue of uncertain disturbances inherent in control algorithms. Reference the ' Robust Integral Sliding Mode Control Design for Stability Enhancement of Under-actuated Quadcopter' and explore potential mitigation strategies in detail.

Conduct a thorough proofreading of the paper to rectify any typos and enhance linguistic clarity.

Ensure that all abbreviations are defined and explained within the text.

Comments on the Quality of English Language

Conduct a thorough proofreading of the paper to rectify any typos and enhance linguistic clarity.

Author Response

(The authors gave the same response as above.)

Round 2

Reviewer 1 Report

Comments and Suggestions for Authors

Thank you for responding to these comments. Your revision has improved the quality of the manuscript. While I am still confused by the design of the study, I am curious to find what you will find in hardware experiments using the control obtained here.

Comments on the Quality of English Language

I think looping in native speakers is not always necessary and can be costly. There are many software or web-based tools that do a pretty good job with grammatical checks. I think even native speakers use them a lot to avoid mistakes in the manuscript. I hope this is a helpful suggestion moving forward.

Reviewer 2 Report

Comments and Suggestions for Authors

Recommended for publication

Comments on the Quality of English Language

Little more effort is needed
